# Isolation and identification of an isoflavone reducing bacterium from feces from a pregnant horse

Xie Jinglong, Li Xiaobin, Zhao Fang, Wang Chenchen, Yang Kailun*

Xinjiang Laboratory of Meat-and Milk-Production Herbivore Nutrition, Xinjiang Agricultural University, Urumqi, Xinjiang, China

☯ These authors contributed equally to this work.
* 1447591725@qq.com

## Abstract

The aim of this research was to isolate bacteria capable of biotransforming daidzein from fresh feces from pregnant horses. A Hungate anaerobic roller tube was used for anaerobic culture. Single colonies were picked at random and incubated with daidzein. High performance liquid chromatography was used to detect whether the isolated bacteria were able to biotransform the substrate. A strain capable of reducing daidzein was selected and characterized using sequence analysis of 16S rDNA, and a phylogenetic tree was constructed. The morphological physiological and biochemical characteristics of the strain were investigated. A facultative anaerobic, Gram-positive bacterium capable of converting daidzein to dihydrodaidzein was isolated and named HXBM408 (MF992210). A BLAST search of HXBM408's 16S rDNA sequence against the GenBank database suggested that the strain has 99% similarity with *Pediococcus acidilactici* strain DSM (NR042057). The morphological, physiological, and biochemical characteristics of HXBM408 are very similar to those of *Pediococcus*. Based on these characteristics, the strain was identified as *Pediococcus acidilactici*. The bacterial strain HXBM408 isolated from the feces of pregnant horses was able to reduce the isoflavone daidzein to dihydrodaidzein.

## Introduction

Dihydro daidzein (DHD) is a metabolite formed by daidzein in the animal intestines through the specific microbial group action. It has higher biological activity than its predecessor of the daidzein, mainly in estrogen-like form. such as estrogen-like effect, antioxidant and enzyme inhibition effect, etc[1–2]. In addition to DHD, after hydrogenation, the ketone and opens the C-ring and so on process. It can be decomposed to produce Dihydodaidzein (DHD), Tihydo-daidzein (THD), Equol, and O-desmethylangolensin (O-Dma), etc[3].DHD and THD have higher biological activity than daidzein. Chin-Dusting and colleagues have found that the ability of DHD and THD to regulate blood vessels is much higher than that of the precursor substance, daidzein[4]. Liang and colleagues studied the free radical scavenging activity of Daidzein, DHD, and O-DMA. The scavenging effect of DHD and O-DMA on free radicals

**Data Availability Statement:** All relevant data are within the paper and its Supporting Information files.

**Funding:** This work was funded by the Ministry of Science and Technology of Xinjiang Uygur

Autonomous Region, Autonomous Region Key Laboratory Project (2016D03012) to YK.

**Competing interests:** The authors have declared that no competing interests exist.

was significantly higher than that of daidzein, probably because the structures of DHD and O-DMA lead to stronger antioxidant activity[5].

There are differences between individual animals. Differences in host genes and diet can produce changes in intestinal flora which can, in turn, affect the degradation of soybean isoflavones. For example, significant differences have been observed in degradation of soy isoflavones in individuals from different geographical regions, and between vegetarians and nonvegetarians[6]. The metabolites of soybean isoflavones have high activity, so improving the efficiency of the degradation of soybean isoflavones may lead to a wide range of potential applications. So, far, researchers have isolated microorganisms which can degrade soybean isoflavones from fresh feces and the intestines of humans, mice, central Jiangsu pigs[7], and other animals The metabolic effects of soy isoflavones were initially identified in the urine of pregnant horses[8], but until now soybean isoflavone degrading bacterial strains have not been isolated from the pregnant horse. In this study we screened bacterial strains from fresh horse manure for their ability to break down daidzein.

## Materials and methods

### Chemicals and instruments

Brain-heart infusion (BHI) and brain-heart infusion agar (BHIA) were obtained from Hopebip (Qingdao). Daidzein and Dihydrodaidzein were sourced from Sigma (St. Louis, Mo.). HPLC-grade acetonitrile and methanol were bought from J.T. Baker (Phillipsburg, N.J.). Carbon dioxide (99.9%) and Nitrogen (99.99%) were obtained from Miquan of Urumqi.

The HPLC system was from Shimadzu (Japan), including a binary gradient elution (LC-20AB), a UV detector (SPD-20A), and a column temperature control box (CTO-lOAS). We also used a Hungate anaerobic culture tube, a DHP-500 electric heating incubator, H-600 transmission electron microscopy, and a MOTIC BA400 biological microscope.

### Culture medium

BHI medium was made from tryptone 10.0g/L, bovine heart leaching powder 17.5g/L, D(+) glucose 2.0g/L, NaCl 5.0g/L, and $Na_2HPO_4$ 2.5g/L. BHIA medium consisted of brain/heart extract, peptone 27.5g/L, D(+) glucose 2.0g/L, NaCl 5.0g/L, $Na_2HPO_4$ 2.5g/L, and agar 15.0/Lg. BHI broth was made using 38g of BHI and 1L of distilled water. Fifty-two grams of BHI agar solid medium was added to 1L of distilled water to produce BHI agar solid medium. One percent methylene blue anaerobic indicator was added, the solution was boiled to remove oxygen, and 0.5% L-cysteine was added. $CO_2$ was passed through the solution for about 30 minutes in a sealed container. The medium was dispensed into Hungate anaerobic tubes using a 5 mL syringe, and incubated. The anaerobic tube was treated with $CO_2$ before packing. The tubes were sterilized at 121˚C for 20 minutes, followed by the addition of daidzein dissolved in DMSO, to a final concentration of 100$\mu$mol/L.

### Bacteria collection and treatment

Sample of fresh feces were obtained from a pregnant *Akhal-teke* horse six years of age, pregnant with her second foal, raised in Xinjiang Ancient Ecological Park. Ten grams of fresh sample was collected in 50 ml of sterile saline and covered with 5 ml of sterilized mineral oil. Four layers of yarn were used to filter bacteria, and after addition of 30% v/v glycerol samples were stored at −70˚C. This study was supported by Xinjiang Agricultural University.

Note:Sample of fresh feces were obtained from a pregnant Akhal-teke horse six years of age, pregnant with her second foal, raised in Xinjiang Ancient Ecological Park.

## Metabolism of isoflavonoids by the fecal cultures

One milliliter of bacterial culture was inoculated into a Hungate anaerobic culture tubes containing 5 mL of the sterilized BHI broth medium with $100\mu$mol/L substrate for the daidzein, and fitted with butyl rubber septa. The anaerobic tube contained 99.9% $CO_2$. The anaerobic tube was placed in the incubator for 48 hours at 37˚C, before samples were taken for HPLC analysis[9].

## Screening bacteria for the metabolic activity of isoflavonoids

The selection of strains was carried out according to the Hungate anaerobic roll tube technique[9]. An aliquot of $100\mu$L of frozen culture was serially diluted using sterile saline in concentrations from $10^{-1}$ to $10^{-10}$. Sterilized BHI solid agar medium was placed in a constant temperature water bath at a temperature of 50˚C. After the temperature stabilized, $100\mu$L of diluted bacteria was inoculated into an anaerobic tube containing 5 mL BHIA. The anaerobic tube was rotated on ice, so that the agar medium solidified in the tube wall. The anaerobic tube was placed in an incubator at 37˚C for 72 hours. Single colonies were transferred to anaerobic tubes containing 5 mL of BHI growth medium supplemented with $100\mu$mol/L daidzein. After 48 hours of incubation at 37˚C samples were taken for HPLC analysis.

## HPLC analysis

One milliliter of each sample was extracted three times with 1 mL diethylether, the ether fractions were combined and then evaporated to dryness under a stream of nitrogen gas and redissolved in $100\mu$L ethanol. The samples were filtered through a $0.4\mu$m filter and stored at 4˚C until analysis[10].

A $20\mu$L sample was separated using a Welch Ulimate C18 column ($250 \times 4.6$ mm, $4\mu$m). The temperature was set to 30˚C and the flow rate as maintained at 0.6 mL/min. Elution was isocratic with a mobile phase consisting of methanol: acetonitrile: water (20:30:50). Daidzein and DHD were detected at 260nm.

## Identification of the strain

**Physiological, biochemical, and morphological characteristics.** The morphology of the strain selected was observed using Gram staining and electron microscopy, and compared with the physiological and biochemical characteristics of the reference strain in Bergey's Manual of Systematic Bacteriology.

**16S rDNA sequence analysis.** The genomic DNA of the strain was extracted using a SK8255 column bacterial genomic DNA extraction kit (Sangon biotech, Shanghai, China). PCR amplification of the 16SrDNA was performed using the universal primers `27F (AG AGTTTGATCCTGGCTCAG)` and `1492R (TACGGCTACCTTGTTACGACTT-3)`. The PCR regime used for amplification was as follows: 94˚C for four minutes, followed by 30 cycles consisting of 94˚C for 45 seconds, 55˚C for 45 seconds, and 72˚C for one minute 30 seconds, and a single final extension step of 72˚C for 10 minutes. The PCR products amplified from the genomic DNA were purified with a SK8131p column DNAJ gum recycling kit (Sangon biotech, Shanghai, China). A pMD®18-T Vector kit (Takara, Japan) was used for ligation and transformation. DNA sequencing was performed by Shanghai Sangon Biological Engineering Technology and Services Co. Ltd. The sequences were compared to known sequences in NCBI, and an unrooted phylogenetic tree of the strains was constructed using the neighbor-joining method in the MEGA version 5.0 software.

## Growth curves and pH change

The strain was grown in 5 mL of BHI liquid medium in an anaerobic tube for 14 hours. An aliquot of $100\mu$L of the culture was inoculated into the fresh BHI liquid medium and placed in an incubator at 37˚C. Samples for the measurement of optical density at 600 nm and pH were taken at 0 hours, 1 hour, 2 hours, 4 hours, 6 hours, 8 hours, 10 hours, 14 hours, 18 hours, 22 hours, and 48 hours.

## Effects of temperature on growth of strain

The strain was grown in 5 mL of BHI liquid medium in an anaerobic tube for 14 hours. An aliquot of $100\mu$L of the bacterial culture was inoculated into fresh BHI liquid medium and cultured at 0˚C, 25˚C, 30˚C, 35˚C, and 40˚C After 14 hours, the optical density of the sample was measured at 600 nm.

# Results

## Bacterial isolation

The bacteria were cultured on an anaerobic roller tube for two days, after which 86 bacterial colonies were isolated and each colony inoculated into 5 mL of BHI medium, and incubated under anoxic conditions. Comparing the uniformity of the single strains after culture with the standard curves of daidzein and DHD, only one strain could be identified as producing DHD as a daidzein metabolite using HPLC. Fig 1 shows the chromatograms of standards for daidzein and DHD, Fig 2 is a chromatogram of the DHD-producing strain. This strain was named HXBM408.

In Table 1, it can be seen that after inoculation of HXBM408 for 48 hours, production of DHD can be detected, but no equol can be detected. In cultures which were not inoculated with HXBM408, the production of DHD and equol were not detected, indicating that strain HXBM408 can degrade daidzein to produce DHD.

## Morphology of the bacterium

Transmission electron microscope images of Gram stained HXBM408 are shown in Fig 3 and Fig 4. Light microscope observations revealed that the strain is a Gram-positive pleomorphic bacillus arranged in several pairs. The bacteria are spherical, $0.5\mu$m to $0.7\mu$m in diameter, and show no flagella or spores.

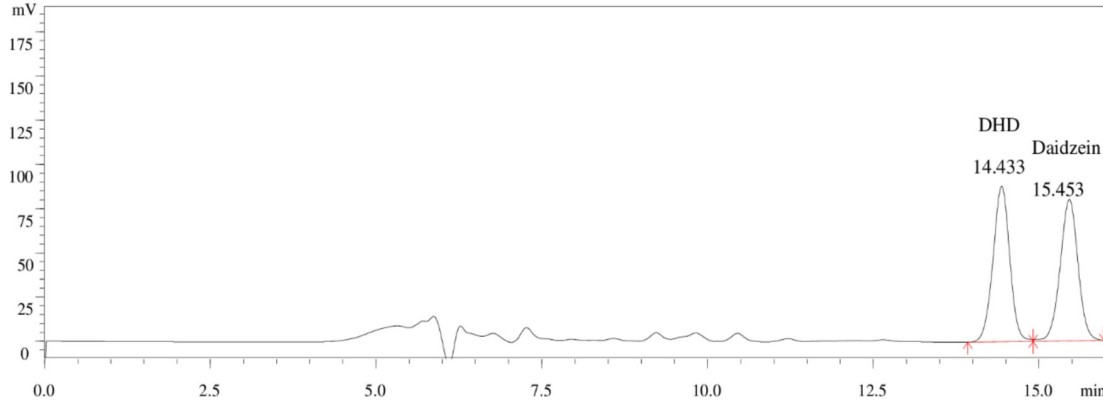

**Fig 1. HPLC chromatogram of standards for daidzein and DHD.**

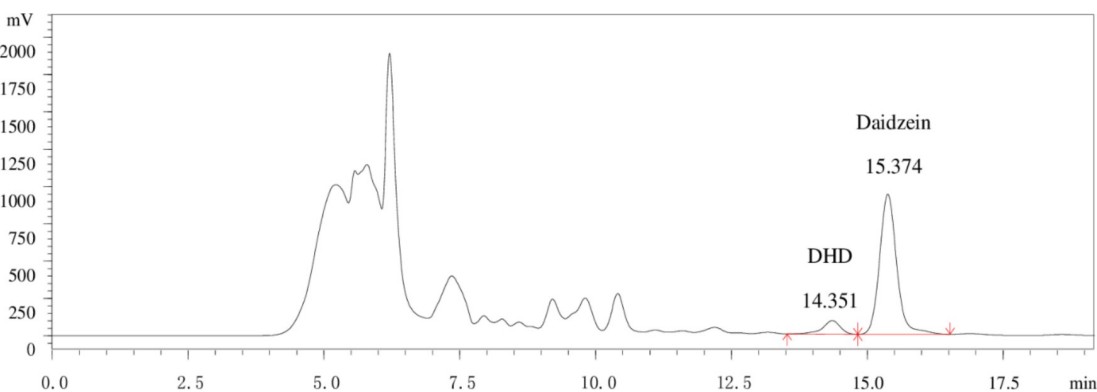

**Fig 2. HPLC chromatogram of the culture medium.**

## Biochemical properties

The morphological and biochemical properties of the DHD-producing bacterium are summarized in Table 2. The strain is facultatively anaerobic, contact enzyme-negative, methyl red-negative, V-P-negative, esculin-negative, and gelatin and arginine dihydrolase-positive. It can utilize other sugars to produce acid, including glucose, raffinose, sorbitol, sucrose, fructose, and lactose. It grows at 45°C, but not at 10°C.

## 16SrDNA sequencing of HXBM408

The phylogenetic tree of strain HXBM408 is shown in Fig 5. We sequenced the 1439-basepair sequence of the 16S rDNA gene of the bacterium. The 16S rDNA nucleotide sequence has been submitted to GenBank under accession no. MF992210. Sequence alignment using the Bioedit software (MEGA6.0) revealed 99% identity of the 16S rRNA sequence of strain HXBM408 with *Pediococcus acidilactici* strain DSM (NR042057).

## Growth curve and pH change

After strain HXBM408 was inoculated into BHI-daidzein medium, the absorbance value of the culture medium was then measured at 600nm with a spectrophotometer, and the pH value was determined using a pH meter (Fig 6). After inoculation, HXBM408 grew rapidly for 10 hours and then leveled out. The pH decreased rapidly until six hours after inoculation, reaching a minimum 6.77, after which the pH value recovered, reaching 6.89 at 48 hours post-inoculation.

**Table 1. Concentration of daidzein and its metabolites in culture medium before and after culture of HXBM408 (μg/mL).**

| Treatment | | Inoculated strain HXBM408 | Uninoculated strain HXBM408 |
|---|---|---|---|
| 0 h | DAI | 20.19±0.03 | 20.11±0.02 |
| | DHD | N/D | N/D |
| | Equol | N/D | N/D |
| 48 h | DAI | 18.86±0.02 | 20.09±0.02 |
| | DHD | 0.11±0.01 | N/D |
| | Equol | N/D | N/D |

ND: not detected

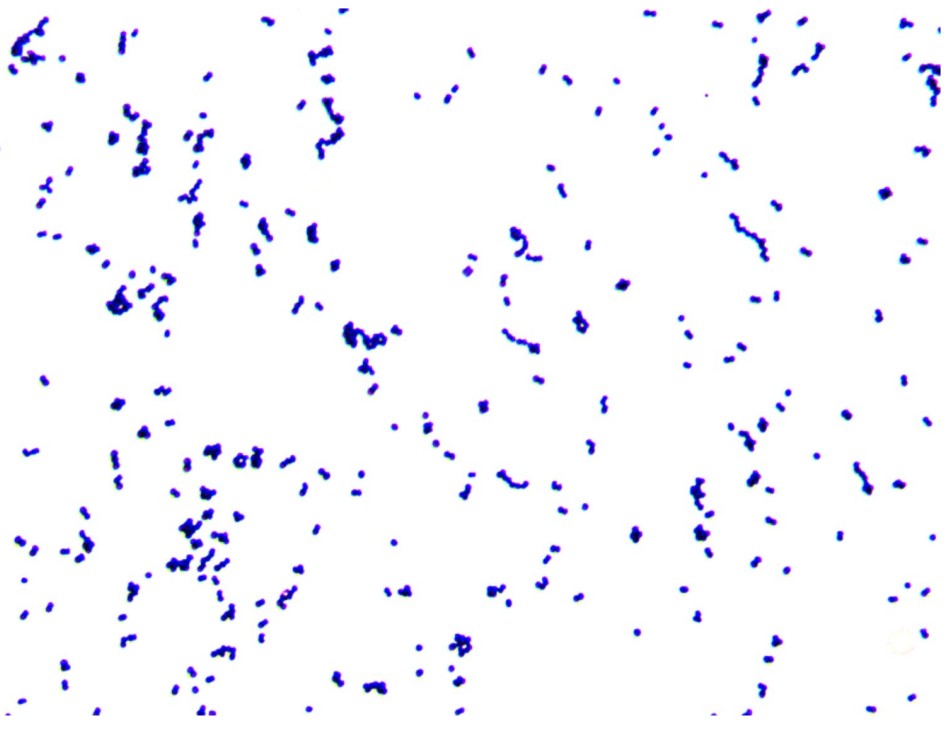

**Fig 3. Gram staining of HXBM408.**

### Effects of temperature on the growth of the strain

The effects of temperature on the growth of the strain are shown in Fig 7. Strain HXBM408 grows well at 32°C to 42°C, with an optimum growth temperature of 37°C. It therefore belongs to the thermophilic microorganisms, and temperatures which are too low or too high will inhibit the growth of the strain.

## Discussion

This experiment used an anaerobic culture of bacteria from fresh feces from a pregnant horse, and used daidzein as a substrate to isolate specific bacterial strains that can produce equol or related intermediates. A new bacterial strain, HXBM408, was isolated using the Hungate anaerobic roller method. This strain does not produce equol, but produces its precursor, DHD. DHD has important functions, apart from its role as a precursor for equol[11–12]. Due to similarities in the structure of DHD and 17β-estradiol, DHD can effectively reduce vasocon-striction, so DHD is one of the active ingredients that maintain the activity of blood vessels to prevent vascular endothelial injury [13–14]. Atkinson et al. found that some antibiotics inhibit the production of equol but have no effect on DHD production[15]. Recently, strains which convert daidzein to DHD have been identified as *Clostridium sp*. HGH6 [16], *Clostridium sp*. TM-40 (AB249652) [17], and Sharpea Niu-O16 (AY263505) [18]. The strain HGH6 and *Clostridium* sp. TM-40 have been isolated from human feces, and strain Niu-O16 was isolated from bovine rumen. The ability of individuals to degrade daidzein can vary by as much as 500–800 times [19]. The strain HXBM408 isolated from this experiment produced a DHD content of 0.11$\mu$g/mL and a low concentration of DHD in the culture medium. In previous studies, it was found that bacterial strains were affected by factors such as temperature, time, and nutrient composition of the medium during the culture process. However, in the present

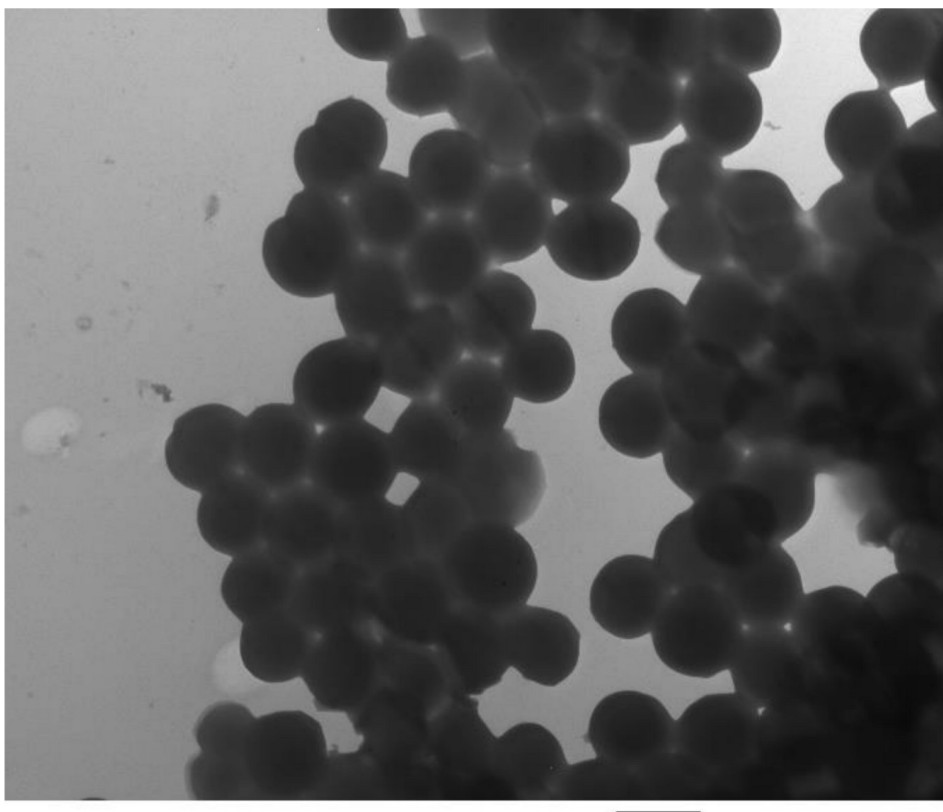

500 nm

**Fig 4. Transmission electron microscope image of HXBM408.**

study we only performed fermentation under fixed culture conditions, in order to establish replicable conditions for future studies. The conditions were intended to maximize the concentration of DHD in the culture fluid.

**Table 2. Physiological and biochemical characteristics of HXBM408.**

| Items | Results |
|:---:|:---:|
| Contact enzyme | - |
| Gelatin | + |
| Methyl red | - |
| V-P | - |
| Arginine dihydrolase | + |
| Esculin | - |
| Glucose | + |
| 10°C | - |
| 45°C | + |
| Raffinose | + |
| Sorbitol | + |
| Sucrose | + |
| Fructose | + |
| Lactose | + |

Note:+: positive reaction; -: negative reaction.

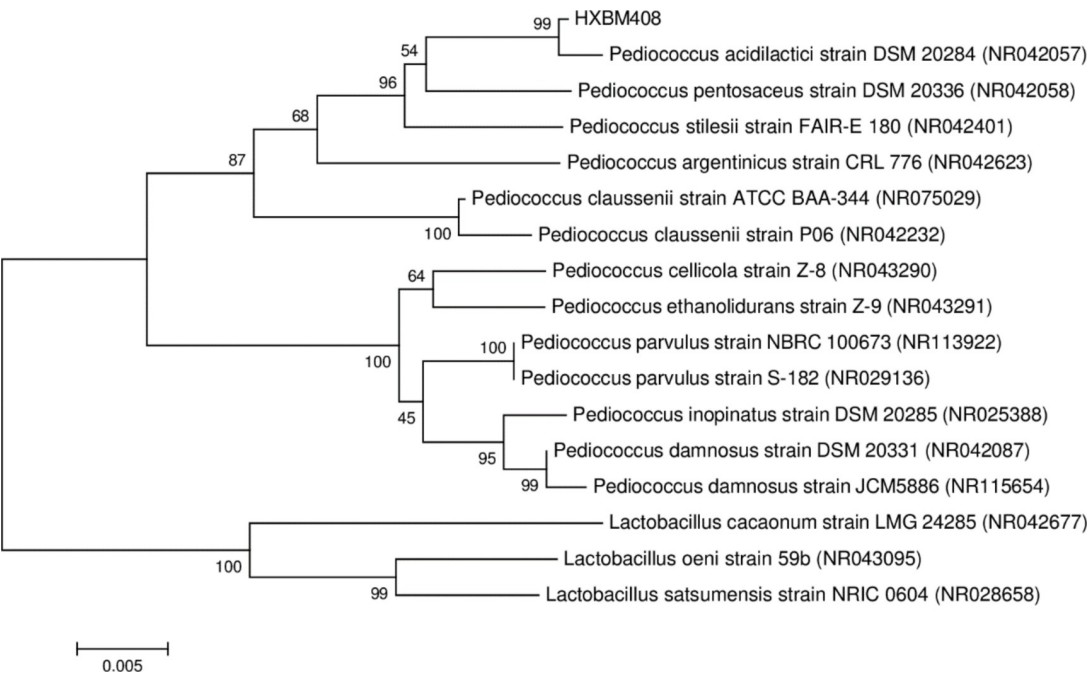

**Fig 5. Phylogenetic tree based on 16S rDNA of strain HXBM408 and reference strains.** Note: Numbers at nodes represent bootstrap values. Numbers in brackets are the accession numbers of sequences in GenBank. The scale bar "0.01" represents sequence divergence.

Intestinal microbes are essential to the degradation of daidzein; sterile animals cannot degrade daidzein[20]. The bacteria reaches a concentration of $10^{14}$ in the animal intestine, but when isolated from the animals the culture of intestinal bacteria which biodegrade daidzein is more complex. At present, the distribution of biodegradation of daidzein has been dispersed, involving *Adlercreutzia* [21], *Bifidobacterium* [22], *Lactococcus* [23], *Streptococcus* [24] *Slackia* [25], and probably more species. Strain HXBM408 has 99% sequence identity of the 16S rRNA gene with that of *Pediococcus acidilactici* strain DSM (NR042057), and the combined

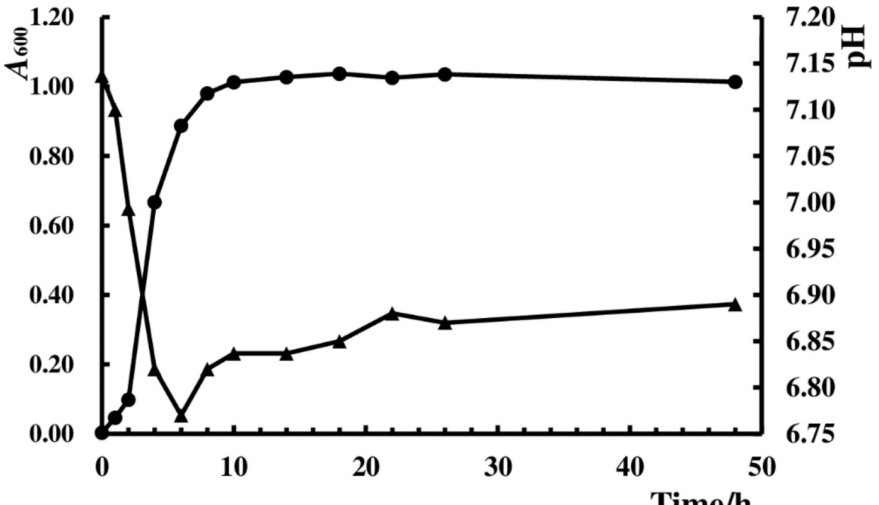

**Fig 6. Growth curve and variation of pH value of strain HXBM408 in BHI-daidzein medium.**

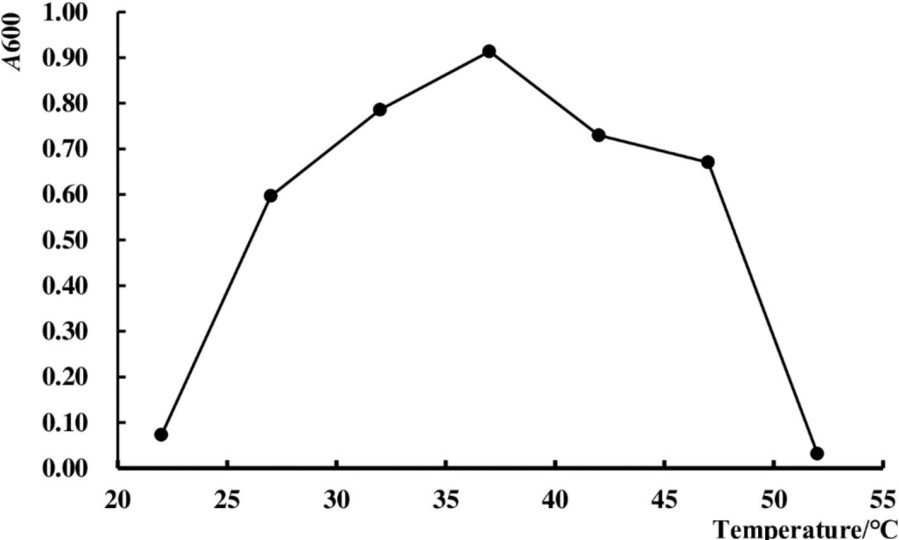

**Fig 7. Effects of temperature on the growth of strain HXBM408.**

morphological and physiological biochemical characteristics of the strain HXBM408 indicate that the strain could be classified as *Pediococcus*.

After strain HXBM408 was inoculated into BHI-daidzein medium it grew rapidly during up to six hours, after which growth leveled off. The pH rapidly decreased over six hours and then stabilized. The growth of the strain was proportional to the pH value of the culture medium, indicating that the strain could be acidified. In addition, temperature plays an important role in the growth of the bacteria, and each strain has an optimum growth temperature; temperature also affects the enzymatic reactions of the bacteria during growth and metabolism. When the temperature was below 22˚C or above 52˚C growth was inhibited, and 37˚C was the optimal growth temperature.

## Conclusions

In this study, strain HXBM408 was isolated from the fresh feces of a pregnant horse. The strain exhibited the ability to produce DHD from daidzein. HXBM408 is classified as *Pediococcus acidilactici*, a facultatively anaerobic, Gram-positive organism with an optimal growth temperature of 37˚C.

## Supporting information

**S1 Table. Concentration of daidzein and its metabolites in culture medium before and after culture of HXBM408.**
(DOCX)

**S2 Table. Growth curve and pH change.**
(DOCX)

**S3 Table. Effects of temperature on the growth of the strain.**
(DOCX)

## Acknowledgments

This work was funded by the Ministry of Science and Technology of Xinjiang Uygur Autonomous Region, Autonomous Region Key Laboratory Project (2016D03012).

## Author Contributions

**Conceptualization:** Xie Jinglong, Wang Chenchen, Yang Kailun.

**Data curation:** Xie Jinglong, Li Xiaobin.

**Formal analysis:** Xie Jinglong, Zhao Fang.

**Funding acquisition:** Yang Kailun.

**Investigation:** Xie Jinglong, Wang Chenchen.

**Methodology:** Xie Jinglong.

**Project administration:** Xie Jinglong.

**Resources:** Xie Jinglong, Li Xiaobin.

**Software:** Xie Jinglong.

**Supervision:** Yang Kailun.

**Writing – original draft:** Xie Jinglong.

**Writing – review & editing:** Xie Jinglong, Yang Kailun.

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
