## [Decision Letter · Decision Letter 0]

7 Aug 2019

PONE-D-19-15247

Isolation and identification of an isoflavone reducing bacterium from feces from a pregnant horse

PLOS ONE

Dear Dr Yang,

Thank you for submitting your manuscript to PLOS ONE. After careful consideration, we feel that it has merit but does not fully meet PLOS ONE’s publication criteria as it currently stands. Therefore, we invite you to submit a revised version of the manuscript that addresses the points raised during the review process.

We would appreciate receiving your revised manuscript by Sep 21 2019 11:59PM. To enhance the reproducibility of your results, we recommend that if applicable you deposit your laboratory protocols in protocols.io, where a protocol can be assigned its own identifier (DOI) such that it can be cited independently in the future. For instructions see: http://journals.plos.org/plosone/s/submission-guidelines#loc-laboratory-protocols

We look forward to receiving your revised manuscript.

Kind regards,

Prakash Kumar Sarangi, PhD

Academic Editor

PLOS ONE

Journal Requirements:

2. Please ensure that you refer to Figures 1, 2 and 7 in your text as, if accepted, production will need this reference to link the reader to the figure.

3. We note you have included a table to which you do not refer in the text of your manuscript. Please ensure that you refer to Table 2 in your text; if accepted, production will need this reference to link the reader to the Table.

Additional Editor Comments:

The anaerobic tube was placed in the incubator for 48 hours at 37°C, before samples were taken for HPLC analysis

Incubation temperature at 37°C : Is not standardized anywhere ( Give the reference)

Some new references inside the text are to be cited as the manuscript has references before 2010/2011.

Reviewers' comments:

Reviewer's Responses to Questions

**Comments to the Author**

1. Is the manuscript technically sound, and do the data support the conclusions?

Reviewer #1: Yes

Reviewer #2: Yes

2. Has the statistical analysis been performed appropriately and rigorously? 

Reviewer #1: I Don't Know

Reviewer #2: No

3. Have the authors made all data underlying the findings in their manuscript fully available?

Reviewer #1: Yes

Reviewer #2: Yes

4. Is the manuscript presented in an intelligible fashion and written in standard English?

Reviewer #1: Yes

Reviewer #2: Yes

5. Review Comments to the Author

Reviewer #1: The manuscript provides imprecise insights in comparison and correlation between faeces of different horse varieties.

Comparison with aerobic bacterial strains is also missing.

Clear evidence with respect to standardization of cultural parameters is not available.

Reviewer #2: The article is written nicely baring few mistakes but discussion part could be improved. The article in my view lacks novelty

6. PLOS authors have the option to publish the peer review history of their article (what does this mean?). If published, this will include your full peer review and any attached files.

Reviewer #1: No

Reviewer #2: No

---

## [Editor Report · Decision Letter 1]

24 Sep 2019

Isolation and identification of an isoflavone reducing bacterium from feces from a pregnant horse

PONE-D-19-15247R1

Dear Dr. Yang,

We are pleased to inform you that your manuscript has been judged scientifically suitable for publication and will be formally accepted for publication once it complies with all outstanding technical requirements.

With kind regards,

Prakash Kumar Sarangi, PhD

Academic Editor

PLOS ONE

Additional Editor Comments (optional):

Paper is accepted.
---

## [Editor Report · Acceptance letter]

29 Oct 2019

PONE-D-19-15247R1 

Isolation and identification of an isoflavone reducing bacterium from feces from a pregnant horse 

Dear Dr. Kailun:

I am pleased to inform you that your manuscript has been deemed suitable for publication in PLOS ONE. Congratulations! Your manuscript is now with our production department. 

With kind regards,

on behalf of

Dr. Prakash Kumar Sarangi 

Academic Editor

PLOS ONE